# Detection of Human p53 In-Vitro Expressed in a Transcription-Translation Cell-Free System by a Novel Conjugate Based on Cadmium Sulphide Nanoparticles

**DOI:** 10.3390/nano10050984

**Published:** 2020-05-21

**Authors:** Víctor Barba-Vicente, María Jesús Almendral Parra, Juan Francisco Boyero-Benito, Carlota Auría-Soro, Pablo Juanes-Velasco, Alicia Landeira-Viñuela, Álvaro Furones-Cuadrado, Ángela-Patricia Hernández, Raúl Manzano-Román, Manuel Fuentes

**Affiliations:** 1Department of Analytical Chemistry, Nutrition and Food Science, Faculty of Chemistry, University of Salamanca, 37008 Salamanca, Spain; victorbarbavicente@gmail.com (V.B.-V.); jfbb@usal.es (J.F.B.-B.); cauriasoro@gmail.com (C.A.-S.); 2Proteomics Unit, Cancer Research Centre (IBMCC/CSIC/USAL/IBSAL), 37007 Salamanca, Spain; pablojuanesvelasco@usal.es (P.J.-V.); alavi29@usal.es (A.L.-V.); rmanzano@usal.es (R.M.-R.); 3Department of Medicine and Cytometry General Service-Nucleus, CIBERONC CB16/12/00400, Cancer Research Centre (IBMCC/CSIC/USAL/IBSAL), 37007 Salamanca, Spain; alvarofucu@usal.es (Á.F.-C.); angytahg@usal.es (Á.-P.H.)

**Keywords:** CdS-BSA quantum dots, nanoparticles, bovine serum albumin, synthesis, fluorescent immunoassays, protein detection, IVTT protein expression

## Abstract

Here, cadmium sulphide quantum dots (CdS QDs) have been synthetized and functionalized with Bovine Serum Albumin (BSA) in a colloidal aqueous solution with a stability of over 3 months. Specific synthesis conditions, in homogeneous phase and at low temperature, have allowed limitation of S^2−^ concentration, hence, as a consequence, there is restricted growth of the nanoparticles (NPs). This fact allows binding with BSA in the most favorable manner for the biomolecule. The presence of Cd^2+^ ions on the surface of the CdS nanoparticle is counteracted by the negatively charged domains of the BSA, resulting in the formation of small NPs, with little tendency for aggregation. Temperature and pH have great influence on the fluorescence characteristics of the synthetized nanoparticles. Working at low temperatures (4 °C) and pH 10–11 have proven the best result as shown by hydrolysis kinetic control of the thioacetamide precursor of S^2−^ ion. Biological activity of the coupled BSA is maintained allowing subsequent bioconjugation with other biomolecules such as antibodies. The chemical conjugation with anti-Glutathione S-transferase (α-GST) antibody, a common tag employed in human recombinant fusion proteins, produces a strong quenching of fluorescence that proves the possibilities of its use in biological labelling. Finally, p53, onco-human recombinant protein (GST tagged in COOH terminus), has been in situ IVTT (in vitro transcription-translation) expressed and efficiently captured by the α-GST-CdS QD conjugate as a proof of the biocompatibility on IVTT systems and the functionality of conjugated antibody.

## 1. Introduction

Quantum dots (QDs) are semiconductor nanomaterials that have attracted considerable attention in several research areas in the last years. Their exclusive optical, electrical and chemical properties make them powerful tools for many applications. Their absorption and luminescence properties can be adjusted from the ultraviolet to the infrared region by changing the size and composition of the QDs [1,2,3]. Over the last 20 years, many strategies have been designed and developed for controlled modifications of QDs to achieve their integration with biological systems [1,2,3]. After the first studies on the use of QDs as labels in the biological experiments carried out by Brucher Jr. et al. or Chan and Nie in 1998 [4,5], the number of biological applications of QDs has exponentially increased, specially related to chemical reactive functional groups (such as primary amines, carboxylic acids, hydroxyl groups or thiols) useful for simple and easy bioconjugations strategies [6]. In addition, QDs have a high specific area, large enough to bind simultaneously multiple biomolecules such as oligonucleotides and peptides [7,8]. In parallel to the successful synthesis of QDs in organic media, several synthesis methods of CdS QDs have been developed in an aqueous medium with high-reproducibility, low cost, ecologically friendly, high solubility and biocompatibility [9]. In the aqueous synthesis of CdS QDs, several chemical moieties (i.e., phosphates or thiols) have been employed as agents to stop precipitation in homogeneous phase; similarly to a hydrophilic polymer, micelles have been also adapted [10]. Commonly, in most biological applications, nanomaterials are coated with proteins (as hydrophilic polymers) to generate biocompatibility and/or adsorbed proteins inherently presented in the biological fluid (i.e., cell/tissue culture or proximal fluid). Thus, in general, the final biological effect of the nanoparticles—in a particular application of interest—is reflected from the effect of nanoparticle per se and the effect of the net protein content [11,12]. Commonly, semiconductor nanocrystals for biological/biomedical applications are synthetized and coated with proteins [13,14]. On the other hand, proteins represent a major source in diagnostic and prognostic biomarkers in biomedical, basic biochemical research and as therapeutic drug targets. Then, among other strategies, fluorescence labeled proteins are useful to explore fundamental biological processes and also employed in a wide variety of biomedical applications [15,16]. However, the preparation of protein-conjugated QDs is a laborious process with multiple stages that usually start with the synthesis of a colloidal QD and its solubilization, and follow by further functionalization with biomolecules. The direct synthesis of protein conjugated QDs under mild conditions (pH, incubation times, T, etc.) has been quite successful in the production of highly luminescent QDs with adequate range of emission wavelengths, high quantum yield and photostability for bioimaging applications. Recently, for this purpose, it has been reported that preparation of QDs enriched with some metallic ions could be combined with some amino acid functional groups (such as carboxylic groups, amino, thiol, etc.) to obtain a covalent bound between chemically reactive moieties of the protein and QD surface [17]. Serum albumins have been used as protein models for different biochemical and physicochemical studies, which include conjugation with nanomaterials, mostly because of their intrinsic function as a carrier biomolecule in biological proximal fluids. Serum albumins are the most abundant proteins in mammalian proximal fluids, such as plasma, where represents about 60% of total protein concentration (approximately 42 g/L). As it is well-known, plasma proteins play a critical role as molecular carriers, mainly because they present binding properties to a wide variety of ligands (i.e., fatty acids, tryptophan, steroids, analgesics, and some fluorescence dyes among others); in fact, the systemic effect in mammalian body of many biomolecules is allowed by the carrier capacity of the plasma proteins for efficient delivery, deposition and accumulation in distal tissues [13,18,19]. As the major soluble protein of the circulatory system, serum albumin has been used as a model protein for many different biophysical, biochemical and physicochemical studies [19]. Bovine serum albumin (BSA) is a very important protein in the blood consisting of 580 amino acid residues with sequence homology to human serum albumin (HSA). It is a versatile carrier protein with wide hydrophobic, hydrophilic, anionic, and cationic properties. Due to its low cost, easy availability, biotechnological relevance and biomedicinal importance, BSA is currently used as a model protein in biomedical research [20]. Aside from these properties, BSA has also been previously reported as a capping/stabilizing agent for the synthesis of several semiconductor nanomaterials based on noble metals [21,22,23,24,25]. However, despite being a very dynamic research area, the use of BSA as ligand in the preparation of colloidal QDs remains relatively unexplored. Here, we explore a novel procedure for preparation and biofunctionalization under different conditions of CdS QDs in an aqueous solution using BSA as a functionalizing coating agent. The novel synthesis procedure being studied aims to obtain BSA-functionalized CdS QDs, with a very small diameter, high fluorescence, and sustained stability over time. Subsequently, the conjugation of the synthesized BSA-QDs with a monoclonal mouse antibody against glutation S-transferase (α-GST) will be tested. This possible novel conjugate, α-GST-BSA-CdS QD, will be employed in the detection of cell-free in vitro GST tagged p53 human protein, expressed by using in vitro human cell- free transcription-translation system (IVTT). IVTT enables not only in situ transcription-translation of human proteins directly from cDNA, but also post-transcriptional and post-translational modifications (PTMs) such as phosphorylation and glycosylation, among others. Compared to other conventional protein expression systems, IVTT can produce functional properly folded proteins of any size in a less complex manner while still being a highly efficient procedure; in fact, it has been adopted as a useful strategy in the Human Proteome Project (www.hupo.org) [26,27].

## 2. Materials and Methods 

### 2.1. Reagents and Solutions

All chemicals used were of analytical grade and were prepared with ultra-high quality deionized water. Bovine Serum Albumin, BSA, (Sigma-Aldrich, www.sigmaaldrich.com, St. Louis. MI, USA). Aqueous solutions of the following reagents, prepared by direct weighing of the species indicated and dissolution in water until the desired concentration was reached: Cd(ClO_4_)_2_·6H_2_O and Thioacetamide, C_2_H_5_NS, 99% (Sigma Aldrich, Darmstadt, Germany), NaOH, KCl, KH_2_PO_4_ and Quinine sulfate (2:1) (salt) dihydrate (Scharlab, Barcelona, Spain): 10^−5^ mol/L solution in 0.5 H_2_SO_4_, NaCl (Panreac, Barcelona, Spain), Na_2_HPO_4_·12H_2_O (Prolabo, Barcelona, Spain, Mouse Monoclonal Antibody Anti-GST (vial 250 µg) Immunostep, SL (Salamanca, Spain). Human IVTT reagents from ThermoFischerScientific. (Waltham, MA, USA). 

### 2.2. Instrumentation

Fluorescence spectra were measured using a Shimadzu (Tokyo Japan) Model RF-5000 spectrofluorophotometer with a Model DR-15 controller unit and a 150 W Xenon lamp as a light source. The UV-visible absorption measurements of the samples were carried out on a Shimadzu UV/Vis-160 spectrophotometer. The pH was measured with a Crison (Barcelona, Spain) GLP 21 pHmeter. Transmission electron microscopy (TEM) was performed on a ZEISS EM-900 device (Oberkochen, Germany). X-ray powder diffraction (XRD) patterns were recorded on a Siemens (Berlin, Germany) D500 X-ray powder diffractometer equipped with graphite monochromatized high-intensity Cu-Kα radiation (λ = 154.050 pm). The refraction index of the samples was determined with an Abbé refractometer (refractometer.com/abbe-refractometer). The infrared spectra (IR) of the samples were measured using a FT-IR Perkin-Elmer Model 1700 spectrometer and the KBr pellets technique with 100 mg of KBr and approximately 0.5 mg of the substance that is being studied.

### 2.3. Synthesis of CdS-BSA QDs in Aqueous Solution

Synthesis of CdS nanoparticles in aqueous solution was as follows: a solution of BSA (3.32 g) in PBS (100 mL, pH = 7.4) was prepared and stirred for 2 h until completely dissolved. 

For pH adjusting, 1 M NaOH was added, to final working conditions: a.-pH 10 and room temperature, b.-pH = 11 and low temperature. Nitrogen was bubbled through the solution in order to avoid the possible oxidation of the S^2−^ ions and this was maintained throughout the experiment. 

With these conditions being maintained, 4 mL of a solution containing the Cd^2+^ 10^−2^ mol/L (from Cd(ClO_4_)_2_·6H_2_O and distilled water) were added and the mixture was maintained in ice. Finally, 2.5 mL of a solution containing C_2_H_5_NS 8 × 10^−3^ mol/L were added and no pH changes was observed. The simplified reactions of the hydrolysis of the precursor and the formation of the nanoparticles are shown in the following equations:(1)$$CH_{3}CSNH_{2}+3OH^{-}\to CH_{3}-COO^{-}+NH_{3}+S^{2-}+H_{2}O$$
(2)$$Cd^{2+}+CH_{3}C\left(S\right)NH_{2}+H_{2}O\to CdS+CH_{3}C\left(O\right)NH_{2}+2H^{+}$$

Both the absorption spectra and the fluorescence excitation-emission spectra were periodically tracked in order to assess the characteristics and stability of the QDs.

### 2.4. Conjugation QD-BSA-α-GST

For the conjugation of QDs and α-GST antibody, both were together incubated in a rotatory shaker for 2 h at 4 °C. Then, 4 different solutions were prepared in which the final antibody concentration ranged from 7.7 × 10^−8^ mol/L to 3.1 × 10^−7^ mol/L and the concentration of QDs was the same in all cases (2.7 × 10^−6^ mol/L). Their fluorescence spectra were recorded in order to study this conjugation. 

### 2.5. Clones and Plasmids

All employed clones, encoding full-length human p53, are from pANT7_cGST clone collection distributed by plasmid repository at Biodesign Institute at Arizona State University (www.dnasu.org). Full-length cDNA clones contain a T7 transcriptional start sequence as well as an internal ribosome entry site (IRES), which is compatible with in vitro transcriptional-translational reagents. In addition, each clone contains an in-frame fused C-terminal GST tag. Each bacterial clone was grown overnight in 5 mL of Luria broth with 100 µg/mL ampicillin. -Plasmid DNA was extracted using Mini-Prep Kit from Promega Inc. (Madison, WI, USA), following manufacture instructions. All plasmids were sequence at Centro de Investigación del Cancer, Genomics Unit (Salamanca, Spain), using a-M13 and T7 primers to confirm the identity of the insert and to ensure that there was not contamination of plasmids stocks. 

### 2.6. Human In Vitro p53 Recombinant Protein IVTT Production

Proteins were synthesized from plasmid DNA using Human In Vitro Protein Expression kit (ThermoFischer, Waltham, MA, USA) following manufacture’s protocol with a few minimal modifications in order to adapt for 1.5 mL eppendorf tubes. Around 1 µg of plasmid DNA was incubated in 25 µL of coupled transcription/translation reaction mix at 30 °C for 90 min. Several amounts of conjugate QD-BSA-α-GST (in DPEC water) were added (final volume 25 µL) to IVTT mix and incubated as described in manufacturer´s protocol. 

Expression vectors (pANT7_cGST) containing an insert corresponding to a protein-coding gene (human p53 is employed here as model) with a glutathione-S-transferase (GST) tag at the carboxyl terminus (Appendix A) were acquired from the human expression clone collection of the DNASU Plasmid Repository at Arizona State University (https://dnasu.org). All clones from the DNASU repository are full-sequence validated as described above. Proteins were synthesized from plasmid cDNA using the 1-Step Human In Vitro Translation kit (Thermo-Pierce, Waltham, MA, USA) according to manufacturer’s protocol. One µg of plasmid DNA was added to 25 µL of reaction mix and incubated at 30 °C for 1.5 h; as it was described above. 

### 2.7. Enrichment of Human GST-Tagged Recombinant Protein Expressed by the IVTT System

To enrich human recombinant GST-fusion human recombinant proteins, 8 µL of slurry (2 µL settled resin) of Pierce Glutathione Magnetic Beads (Thermo-Pierce, Waltham, USA) were used per sample, equilibrated 3 times with 25 µL slurry volume phosphate-buffered saline (PBS), and re-suspended in 12.5 µL slurry volume with PBS at room temperature. A 100 µL aliquot of the equilibrated slurry was added to each well (2 mL). Microbeads were pulled down by a 96-well plate magnet and supernatant was removed. Completed IVTT reaction solution was added to the beads and the bead-protein mixture was shaken at 550 rpm on a plate mixer at 4 °C overnight.

The supernatant was removed, and then 100 µL of wash buffer was added to the beads. The beads were stirring at 550 rpm for 5 min. Six washes were performed in this manner; two with PBS, followed by two with PBS supplemented with 863 mM NaCl, and two with 100 µL PBS supplemented 50 mM ammonium bicarbonate (pH 7.8). After the last wash, beads were re-suspended in 100 µL PBS with 50 mM ammonium bicarbonate (pH 7.8) and 0.05% PPS silent surfactant (Expedeon, San Diego, CA, USA) and stored at −20 °C until digestion. For gel electrophoresis of enriched protein samples, 5 µL of the final resin-slurry was added to 2.5 µL PBS and 7.5 µL Laemmli sample buffer (Bio-Rad, Hercules, CA, USA) and run under reducing conditions. The gel was rinsed with water three times for 5 minutes with mild stirring and stained in SimplyBlue Coomassie stain (Life Technologies, Carlsbad, CA, USA) for 30 min with mild stirring. The gel was destained in water overnight, with rocking, and imaged with an Alpha Innotech Imager System

### 2.8. SDS-PAGE and Immunoblotting

Undigested protein extract from each of fractions was boiled in 1x LOS buffer and separated on a 4–12% bis-Tris denaturing and reducing SDS-PAGE (Invitrogen, Carlsbad, CA, USA). Gels were then subjected to either silver staining (Invitrogen) or transferred onto a nitrocellulose membrane (Bio-Rad, Hercules, CA, USA) for immunoblotting. Membranes were blocked with 5% non-fat dry milk (Safeway, Pleasanton, CA, USA) in TBS-Tween buffer and probed for schistosomal GST (GE 27-4577-01). All primary incubations were done at 4 °C (overnight using a 1:1000 dilution (*v*/*v*). Secondary incubations were performed in 5% non-fat dry milk in TBS-Tween using 1:10.000 (*v*/*v*) diluted peroxidase-conjugated rabbit anti-goat IgG (H+L) (Thermo-Pierce, Waltham, MA, USA). Membranes were visualized using an ECL Plus Western Blotting kit (Amersham plc, Amersham, UK) and detected with radiographic film (ThermoFischer, Waltham, MA, USA).

## 3. Results and Discussion

### 3.1. Physical and Chemical Characterizations of QDs

#### 3.1.1. Absorption and Fluorescence Spectra

The spectrum showed an absorption band of below 400 nm with a maximum of 338.5 nm (Figure 1). The appearance of this band was due to the presence of CdS QDs, in which the electrons were quantically confined and thus able to absorb electromagnetic waves of a given energy and to carry out electronic transitions through the empty bands of the semiconductor. 

The maximum of the absorption band of the QDs could be seen to be well-defined as their amount in solution increases and their size distribution around the mean value decreases. The value of absorbance at 338.5 nm is A = 1.134; at that wavelength, it is proportional to the concentration of nanoparticles in solution although the absorption coefficient depends on the nanoparticle diameter. For wavelengths lower than 300 nm, a non-specific band was observed due to more energetic electronic transitions. It is possible to determinate the mean value of the QDs’ diameter from the wavelength of the absorption maximum (Appendix A). In the present case the diameter was 2.78 nm (Figure 1).

The emission wavelength depends on the NPs’ diameter and the intensity of the emission process depends on the condition of their surface (Figure 2). Once the optimum λ_ex_ had been determined (Figure 2a), the emission spectrum was tracked (Figure 2b) obtaining a maximum emission intensity at 496.0 nm with a value of PL intensity of 381.63 (au). The symmetrical shape of the obtained emission spectrum indicates that the nanoparticles’ surface was highly homogeneous and with no major imperfections.

On the other hand, FWHM (Full Width at Half Maximum, Δ), indicates the greater or lesser size distribution of the nanoparticles around the medium diameter. There is a direct correlation with the FWHM and the size distrubution, because the lower value (a narrower emission band), then a smaller the size distribution (Gaussian, smaller standard deviation) [28]. In Figure 2a, Δ = 104 nm was obtained. This value, Δ, will allow a comparison between the size distribution for different experimental conditions and different lifetimes of the nanoparticles. The Δ value for nanoparticles obtained in organic medium ranges from 30 to 90 nm. Other parameters of these novel QDs that could be studied, such as the number of CdS molecules contained and ε molar extinction coefficient, are properly described in Appendix A.

#### 3.1.2. Effect of the pH in the Medium on the Synthesis of CdS-BSA QDs.

From all the variables that could affect the hydrolysis reaction of thioacetamide in order to form the sulfide ion (S^2−^), pH of the aqueous solution is one of the most important before the addition of the precursors. THe absorption process was started at wavelength values bellow 515 nm (λ of the initiation of the absorption of the unit cell of the CdS crystal), indicating that the small CdS nanoparticles are obtained and the displacement refers to the quantum confinement.

When synthesis was carried out at pH = 9.0 (Figure 3), the maximum absorption is increased, but at t =13 days turbidity in the solution begins to be observed. At pH = 10.0, a higher absorbance is obtained which indicates a higher speed for NPs formation. This effect could be attributed to the greater speed of thioacetamide hydrolysis. Under these conditions, the NPs remain stable during 28 days. However, higher pH values present lower absorbance signals and therefore, higher solution instability (Figure 3a). These facts indicate that in homogeneous phase synthesis method, the concentration of S^2−^ ion is directly influenced by the pH of the solution. 

On the other hand, the chemical process of the appearance of hydroxylated species must also be considered. The Cd^2+^ ions, which are in excess compared with S^2−^ ions, may take part in other reactions such as the formation of hydrated oxides which will be deposited on the surface of the nanoparticles, hence this explains why turbidity is not observed in the solution during days. Then, there is an indication that no cadmium hydroxide precipitate Cd(OH)_2_ is in the solution. 

The initial pH differences will essentially affect the starting availability of S^2−^ concentration to form the nanonuclei, together with the possible formation of a nanoshell of Cd(OH)_2_, which we have already observed in other QD synthesis procedures. These previous studies also demonstrated that the process will be more thermodynamically favourable at higher pH values [29].

The formation of this nanoshell has a considerable effect on the fluorescence of the resulting nanoparticles. The chemical process leads to nanoparticles that maintain their size and surface modificaton, which also might affect the radiation absorption processes. This phenomenon may be positive when the surface alteration is minimal and will be negative when the size of deposit exceeds a critical value.

The behaviour of the BSA as coating agent -under different pH conditions during synthesis- must also be taken into consideration. BSA has accesible several chemical moieties –SH, –NH, –OH and other chemictal groups which have excellent adsorbability and can sequester, entrap and interact with Cd^2+^ [24]. There is a certain chelation time to complete the reaction between BSA and Cd^2+^.

Under the experimental conditions of this study, pH between 9 and 11.0, the NPs obtained throughout present small differences during the evolution process varying their diameter from 2.71 to 3.26 nm. The number of CdS molecules contained in their interior ranges from N = 206 to N = 349. The size of the nanoparticles explains the fact that all their solutions are colourless (Appendix A).

The turbidity observed in the evolution process may be related to the formation of intermediate species (between the Cd^2+^ and the BSA) and their solubility. In an alkaline medium, moslty of the proteins have a negative net charge; thus the charged metallic ions can be bound to the protein molecules to precipitate metal proteinates. This tendency of precipitating proteins is the basis of the toxicity attributed to ions such as Cd^2+^, which has affinity with the carboxylate and sulfhydryl groups of many proteins [24,30,31]. The appareance of these species is in competition with the CdS nanoparticles formation’s reaction.

As the generation speed of the S^2−^ ion by the thioacetamide hydrolysis is greater at pH 10, this condition would be the best for the stability of QDs (Figure 3b).

Different solutions of QDs at different pH were tested for their stability. From the point of view of the fluorescence intensity, at pH 7.0 and 8.0, very low and almost identical fluorescence intensities values are observed. As from pH 9.0 a significant increment of IF_max_ can be observed, which evolves from blue to yellow after 13 days. When the appearance of turbidity begins to be noticeable and the fluorescence intensity decreases.

At pH 10 the IF_max_ of the sysnthetized nanoparticles increases during the first 15 days of evolution, which is probably due to an increament in their concentration together with the possible formation of the nanocrust of Cd(OH)_2_, confirming that it is more favourable at pH 10. This pH is practically constant during 28 days.

However, at pH of 10.5 and 11.0, the value of IF_max_ and the stability of the solution decrease. These results also confirm that shape and surface state depend on the pH, which has an influence not only on kinetic aspects of nucleation and growth but also on the formation of the nanoshell of Cd(OH)_2_. The decrease in fluorescence could confirm the hypothesis of the relationship between the latter and the nanoshell of Cd(OH)_2_ of a specific size deposited on the surface of the QD.

In this study, the results provide an evidence that CdS QDs are stabilised by BSA in a colloidal aqueous solution over a long period of time in the most favourable conditions. This stabilization could be a consequence of the homogeneous synthesis conditions in which the thermodynamic and kinetic aspects (associated with the formation of CdS-BSA) are highly favourable. The pH variation allows the control of the concentration of S^2−^ ions in solution, limiting the possibility of the instantaneous growth of the NPs and allowing the appropriate induced 3D shaping of the BSA, and subsequently the NPs [13]. The presence of Cd^2+^ ions on the CdS NPs surface is counteracted by the negative charge areas of the BSA. The evolution in NPs size appears to indicate that the growth is due to an Ostwald ripening process (Appendix A).

#### 3.1.3. Effect of the pH of the Medium on the Synthesis of CdS QDs at 4 °C

##### Influence of Temperature

Temperature is a critical parameter in the formation of nanocrystal and also in the hydrolysis reaction of thioacetamide, because it plays an important role in the study of CdS. In this study, among the studied synthesis methods [26], it was observed that better synthesis yields are achieved at low temperature. In these cases, small CdS nanoparticles are obtained with high stability and fluorescence. A study was carried out in order to determine the influence of this variable on the characteristics of the CdS QDs at adjusting pH to 9, 10 and 11 values and different concentrations of thioacetamide and BSA.

After addition of the Cd^2+^ precursor, solutions were cooled in an ice bath until the temperature reached 4 °C. These solutions were kept at 4 °C in a refrigerator. From t = 0, aliquots were taken periodically to record their absorption spectra and their luminescence spectra (Figure 4). At low temperature and pH 9 or pH 10, the thioacetamide hydrolysis reaction is slowed in generating sulphur ions. Hence, a low concentration is generated and the nanoparticles form more slowly with smaller diameters (Appendix A). At 4 °C, an incubation of 35 days is required to observe the formation of nanoparticles with a size approx. 3 nm.

At pH 11, during 3 days of incubation, highly stable nanoparticles are also obtained. The mean radius is constant and of ca. 1.5 nm is maintained for at least 82 days. The number of molecules per nanoparticle varies very slightly between 282 and 300. No significant changes occur to the λ_max_ absorption during all the studied period (nearly 3 months). A CdS QD coated with BSA has been obtained with a very small size and high stability over time (Appendix A). At pH 11 under the optimum conditions, the nanoparticles do not grow according to the Ostwald model; there is rather a stable dispersal over time of small NPs.

The fluorescence intensity at maximum emission increased with the pH (Figure 4b) and the best results were achieved at pH 11. In all cases, it was initially increased and constant after a certain evolution time which is different for each pH. It is interesting to note the evolution of the size of the NPs, which is maintained practically constant during all this time; due to the optimum slow hydrolysis for the formation of the nanoparticles of thioacetamide at this pH and 4 °C. The size distribution around the mean value Δ is kept in the 93–108 nm range under optimal conditions (Appendix A).

This indicates that the shape and surface status are highly influenced by the solution conditions such as pH and the temperature; because they are mainly led by kinetic aspects of nucleation and growth and the formation of a nanoshell of Cd(OH)_2_. Once again, it is confirmed that there must be an optimum size for this nanoshell related to the radiant and no radiant emission processes. This nanoshell seems much more homogeneous during work at low temperatures.

##### Influence of the Molar Cd^2+^/thioacetamide Ratio at 4 °C

The concentration of S^2−^ precursor (thioacetamide) could produce a significant influence on the synthesis process of CdS-BSA nanoparticles when it takes place in a homogeneous phase at low temperature. Then, four different solutions were prepared with a final concentration of thioacetamide that varied between 1.9 × 10^−4^ mol/L and 1.9 × 10^−3^ mol/L ([Cd^2+^]/[thioacetamide] (a ratio of between 4.0 and 0.4). Fluorescent emission spectra at the optimum wavelength were analyzed for all of them (Figure 5).

The fluorescence intensity at maximum emission evolved over time in all the experiments. For higher values of the [Cd^2+^]_F_/[thioacetamide]_F_ ratio, high values of fluorescence intensity were initially achieved. For lower ratio values of the nanoparticles showed lower fluorescence intensities. As for the diameters of the nanoparticles synthesized; although wide variations are not observed neither with the concentration of the ion sulphur precursor nor with the time for each concentration, these are smaller for the lowest values of the thioacetamide concentration studied (Appendix A).

From a practical point of view, it seems that best conditions are obtained when a [Cd^2+^]_F_/[thioacetamide]_F_ ratio of 1–2. A quantum yield of 22% was determined for the QDs solution under these conditions (Appendix A) similar to the ones reported for QDs of CdS synthesized in aqueous medium by other procedures [32].

##### Influence of the Molar Cd^2+^/BSA Ratio at 4 °C

For this study, 4 solutions were prepared with a final BSA concentration of between 1.9 × 10^−4^ and a 1.5 × 10^−3^ mol/L ([Cd^2+^]/[BSA] ratio of between 4.0 and 0.5). The evolution of the absorbance values and the diameters of the synthesized NPs for the different relations are displayed in Appendix A. In all cases, fluorescent emission spectra were recorded at the optimum wavelength (Figure 6). The maximum fluorescence intensity evolved in a similar way in the 4 cases although higher intensities were found at ration 2 of [Cd^2+^]_F_/[BSA]_F_.

### 3.2. Characterization of the CdS QDs

The shape of the CdS QDs was studied with Transmission Electron Microscopy (TEM) after evaporating off the solvent (Figure 7a). The QDs morphology was also studied by X-ray diffraction (Figure 7b). The X-ray diffractogram (XRD) has been obtained using as monochromatic radiation of λ = 1.54178 Å (Cu-Kα). Distances have been calculated using Bragg’s law, λ = 2dsin (θ), (symbols appear in Figure 7b).

In Figure 7, it is displayed the diffractogram and diffraction angles, the corresponding distances between crystallographic planes and the corresponding Miller indices (notation system in crystallography for planes in crystal (Bravais) lattices), hkl for the zinc blende structure for the CdS. The CdS can crystallize as ZnS wurzite (hexagonal) or as ZnS blende (face centered cubic). The peaks are well-defined which indicates a high crystallinity of the sample. The obtainment of one or the other phase depends on the method of preparation [33,34,35,36]. Although the diffractograms for both structures show almost coincident peaks; the peak at 31.8º is indicative of the cubic or zinc blende structure [33]. This peak appears with an overlapping shoulder with the closest peak (low intensity) [34,37]. XRD patterns for samples of nanoparticles having different sizes and shapes can look different, and careful analysis of the XRD data can provide useful information and also help correlate microscopic observations with the bulk sample [34].

Infrared spectroscopy was used to assess the binding between BSA and CdS (Figure 7c). As can be observed, in both spectra of the nanoparticles the main bands of the albumin appear, which indicates that a there has been binding (or coating) between the CdS and the BSA. This binding will not be strong (covalent or ionic) as no appreciable displacement of the bands could be observed. It is rather a weak binding of the Van der Waals type, hydrogen links, or electrostatic interactions. 

### 3.3. Conjugation QD-BSA-α-GST

In order to study the fluorescence changes of the synthetized QD-BSA in its conjugation with the α-GST antibody, 4 different solutions were prepared in which the final antibody concentration ranged from 7.7 × 10^−8^ to 3.1 × 10^−7^ mol/L and the same concentration of QDs in all cases (2.7 × 10^−6^ mol/L). Their fluorescence spectra were recorded in order to determine the conjugation. (Figure 8 and Appendix A).

A strong interaction occurs between the QD-BSA and the antibody, which produces a significant fluorescence quenching to allow its use in the other goal of this work, making QDs biological markers, as it can be observed in the next experiments.

### 3.4. Human Recombinant Protein IVTT Expressed

Human IVTT is a cell-free translation and transcriptional system to in situ synthesizes proteins within any size in a highly efficient manner [27]. Another advantage of this system is the production of properly folded proteins with a similar activity as proteins produced by conventional mammalian and/or bacterial expression strategies but with no use of any cell as a fully complete system for the synthesis.

First, it should be noted that the protocol used for protein expression is focused on the production of recombinant proteins from a clone (Appendix A). Recombinant fusion proteins are polypeptides originated by translation of two or more genes previously linked in reading frame to give rise to a single protein. When it comes to detecting a fusion protein, the simplest thing is to use them fused to other proteins, acting as protein markers or tags, by means of the amino- or carboxyl-terminal end, thus preventing inclusion bodies from forming, improving the protein folding and reducing proteolytic degradation [38]. One of the most popular proteins for this role is glutathione S-transferase (GST) [39,40]. In this work, the fusion protein was p53 linked to the C ’terminal end of the GST tag, so that the reading frame from the N′ terminal end to the C′ terminal of the obtained fusion protein was facilitated; and detecting the presence of the GST tag would ensure that p53 protein has been also previously expressed.

As QD-BSA has been successfully conjugated with α-GST antibody, it was possible to carry out one of the main objectives of this work, the evaluation of the cytotoxicity of CdS-BSA NPs in a cell-free system, using an in vitro expression system for human recombinant proteins.

For this objective, there are two possibilities to successfully detect the IVTT expressed protein, either detect GST domain that serves as a control of the correct full-length protein expression or detect p53 domain, which is the protein of interest. Both proteins are included in the recombinant fusion protein as displayed in Figure 9.

Results obtained for the expression of the fusion human protein p53-GST in the presence of QD-BSA-α-GST synthesized in the cell-free transcription-translation system in Western Blot are displayed in Figure 10. They compare the detection of the protein tag, GST, and the protein of interest, p53, at different concentrations of QDs (dilution 1:1 (*v*/*v*)—dilution 1:2000 (*v*/*v*)), volumes of IVTT mix (5 µL and 25 µL) and cDNA encoding the human p53 onco-protein (1 µg).

All results of Western Blots show bands between 100 and 130 KDa, what agrees to the expression of p53 and GST in a fusion protein within the IVTT. Taking account conditions used for experiments there are several considerations that should be highlighted for further protocols. Attending to dilutions, from 1:10 (*v*/*v*), the second lowest used for experiments, it was possible to detect the GST tag and p53 protein. This is an interesting result that shows the sensitivity of our protocol. Related to volumes of IVTT assayed, the lowest one in the protocol (5 µL) was enough to detect both proteins as it could be observed in the Western Blots (Figure 10a,b). Wavelengths used for the final detection have also differences. As it is displayed in the Figure 10, observed bands -at molecular weight that agree with fusion protein- are detected in both cases, however the wavelength of 480 nm is the one that shows more defined results for tested conditions in comparison with 680 nm (Figure 10a,b) (Appendix A). This analysis confirms the successful of the IVTT and the possibility of using this cell-free translation and transcriptional system to synthesize proteins coupling with the QD-BSA at the same time that contributes with the optimization of the protocol adjusting concentrations of reagents and the way of final detection.

### 3.5. Characterization of the QD-GST-p53 Conjugated

An important point is to decipher if the expressed protein conjugated maintains their optical properties in order to be use its fluorescence as a tool for further applications of the system.

Therefore, in order to evaluate the characteristic optical properties of QDs after protein expression, it was decided to perform an experiment based on fluorescence spectroscopy. Previously in this work, we have determined the fluorescence of the QD-BSA-α-GST, noting that the antibody attached to the QD undergoes a fluorescence quenching in the complex (Figure 8). Considering these results, we carried out an experiment based on paramagnetic microspheres, a micro-sized system formed by methacrylate coated with gamma-magnetite nanoparticles (super-paramagnetic properties).

These microspheres are coated with glutathione affinity groups that react with the active site of the GST protein through an enzyme-substrate reaction, capturing the QDs that have expressed the proteins, forming a multi-complex represented in Figure 11.

With this reaction, it is confirmed that IVTT expressed protein is presented in the complex analysis the complex studies, and not only the QD-BSA-α-GST conjugate. The results obtained in the fluorescence experiments are displayed in Figure 12, where different fluorescence is observed between the glutathione affinity microspheres as a negative control and the microspheres with the QDs.

First spectrum represents the behavior of the microspheres as a negative control (Figure 12a). As it can be observed, there is no fluorescence signal apart from the maximum obtained by the solvent. Second spectrum corresponds to the fluorescence related to the conjugate QDs-BSA-α-GST used as a positive control (Figure 12b). The result of the spectrum accords with the results obtained previously in this work (Figure 8) and determined a maximum emission of 440 nm for the complex. The last spectrum corresponds to the final conjugate with the IVTT expressed protein that also shows a maximum of 440 nm as occurs in the positive control (Figure 12c). Even though the intensity of the signal is lower compared with the complex without protein, the fluorescent properties of the QDs remain even after they have been used in IVTT systems.

This fluorescence analysis is extremely important for the further purposes of the QDs as a platform for IVTT. As the IVTT system has already been successfully employed for protein-protein interaction studies [38] and for elucidating intracellular pathways of novel compounds [41,42], the biocompatibility with QDs opens new perspectives for these NPs in biology and medicine applications related with detection of diagnosis proteins thanks to their chemical and physical properties. At the same time, these results confirm the potential of in vitro cell-free systems for in situ expression of proteins in different contexts such as expression oncogenic proteins, like p53 detected in this work, that could be a great enlargement.

## 4. Conclusions

Both at room temperature and at a low temperature (4 °C) the mixture of aqueous solutions of Cd(ClO_4_)_2_·6H_2_O and C_2_H_5_NS in the presence of BSA as a stabilizer leads to the formation of a solution containing CdS nanoparticles with a mean size of below 5 nm, which is confirmed by the absorption spectra and TEM. According to the obtained X-ray diffractograms, the CdS NPs crystallized on the same cubic network corresponding to the zinc blende. No other crystalline phases were observed in their structure.

The experimental synthesis conditions in the homogeneous phase by using thioacetamide as a precursor allowed the coupling with BSA in the most favorable way for the biomolecule. The presence of Cd^2+^ ions on the surface of the nanoparticles is counteracted by the negatively charged moieties of the BSA. This is causing the formation of small nanoparticles which were stable for long periods of time with a low aggregation tendency.

Both the pH and the temperature are variables with a strong influence on the luminescent characteristics of the CdS nanoparticles. The best results were achieved at working conditions of low temperatures (4 °C) and pH range of 10–11. Under these conditions, the kinetic control of the thioacetamide hydrolysis allowed the NPs to show high fluorescence intensities sustained over time and stability within 3 months.

The fluorescence of the resulting CdS QDs is long-lasting enough for them to be used as tracers for inorganic or organic species that interact with them. The coupled BSA to the CdS QD maintains its biological activity that is evaluated by the conjugation with the antibody α-GST, which produces a significant quenching of fluorescence.

The combination of IVTT with CdS QDs has been successfully demonstrated. QDs-BSA-α-GST are biocompatible with the expression of oncogenic human proteins, e.g., p53 and it opens a new perspective that could be used to express a huge number of human recombinant proteins. Moreover, QDs-BSA-α-GST maintain their characteristic optical properties even after being incubated with IVTT, which contributes to the establishment of this platform for the detection of protein-protein interactions and the in vitro synthesis of human recombinant proteins with clinical and biomedical interest.

## Figures and Tables

**Figure 1 nanomaterials-10-00984-f001:**
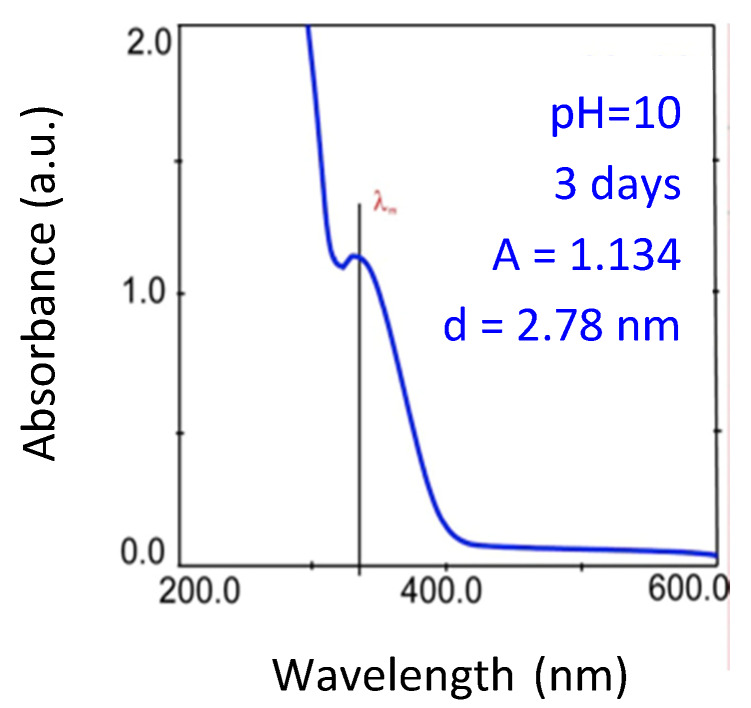
Absorption spectrum of the cadmium sulphide-Bovine Serum Albumin (BSA) quantum dots (CdS BSA QDs) 3 days after the synthesis process: pH = 10; Temperature: 22 °C (“A” is absorbance and “d” is diameter).

**Figure 2 nanomaterials-10-00984-f002:**
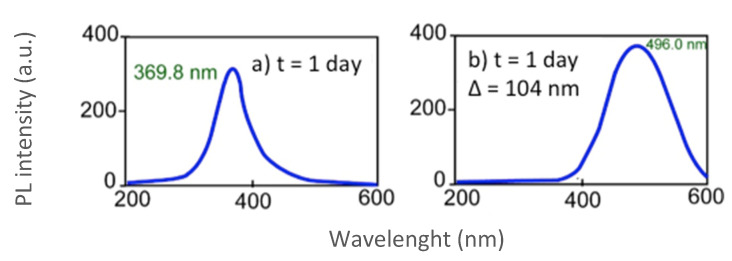
Photoluminescence behavior of the CdS-BSA QDs 1 day after synthesis under the general conditions described. Excitation and emission slit widths R_ex_= R_em_ = 10 nm. (**a**) Excitation spectrum at λ_em_ 496.0 nm. (**b**) Emission spectrum at λ_ex_ 369.8 nm.

**Figure 3 nanomaterials-10-00984-f003:**
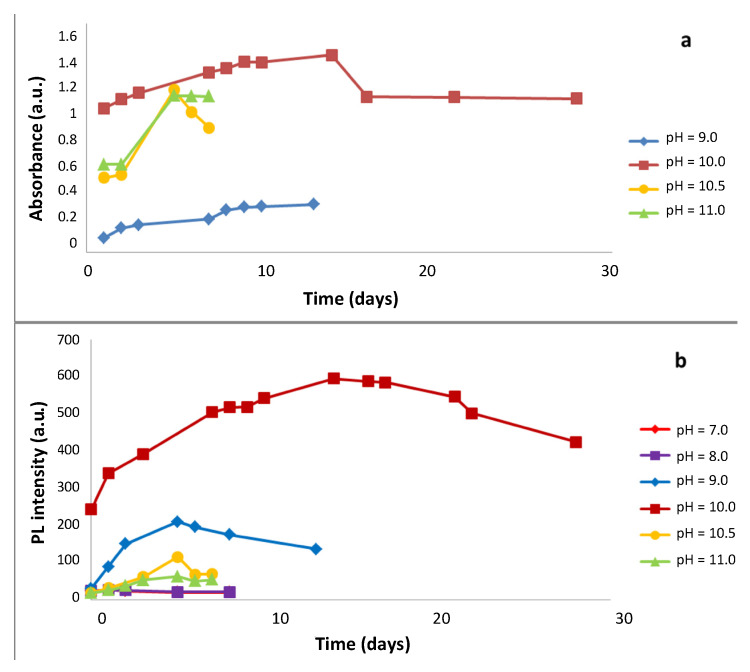
Influence of the change in pH during synthesis. (**a**) Evolution of the maximum absorbance values at different times for different pH values. (**b**) Temporal evolution of the Photoluminescence intensity at different pH values. Under operating conditions the final concentration and their molar ratios were as follows: [Cd^2+^]_F_ = 7.477 × 10^−4^ mol/L; [BSA]_F_ = 3.738 × 10^−4^ mol/L; [thioacetamide]_F_ = 3.738 × 10^−4^ mol/L; [Cd^2+^]_F_/[thioacetamide]_F_ = 2.0; [Cd^2+^]_F_/[BSA]_F_ = 2.0, Room temperature.

**Figure 4 nanomaterials-10-00984-f004:**
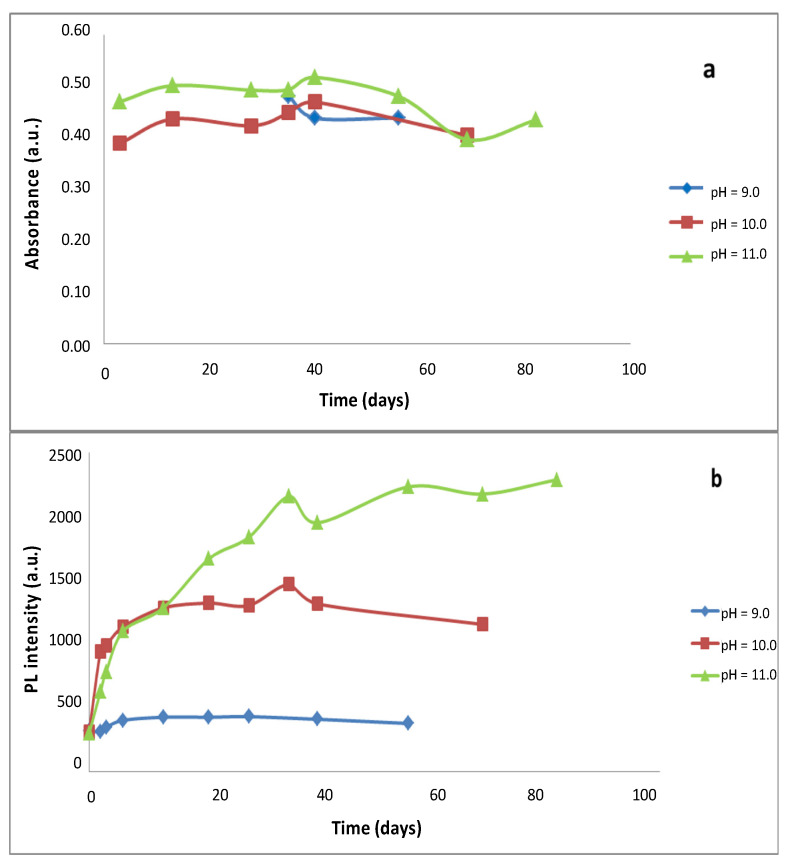
Influence of the change in pH values during synthesis at low temperatures. (**a**) Evolution of the absorbance values of the nanoparticles at different times for different pH values. (**b**) Temporal evolution of the Photoluminescence intensity at different pH values. Under operating conditions the final concentration and their molar ratios were as follows: [Cd^2+^]_F_ = 7.477 × 10^−4^ mol/L; [BSA]_F_ = 3.738 × 10^−4^ mol/L; [thioacetamide]_F_ = 3.738 × 10^−4^ mol/L; [Cd^2+^]_F_/[thioacetamide]_F_ = 2.0; [Cd^2+^]_F_/[BSA]_F_ = 2.0, 4 °C.

**Figure 5 nanomaterials-10-00984-f005:**
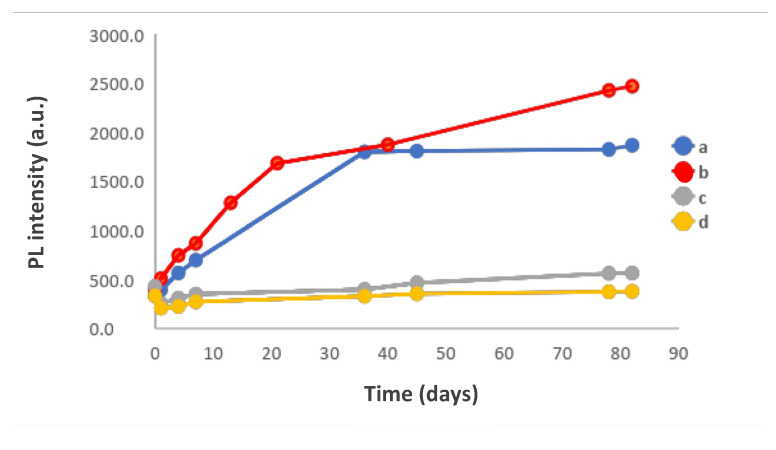
Influence of [Cd^2+^]_F_/[thioacetamide]_F_ at 4 °C. Temporal evolution of Photoluminescence intensity of CdS QDs solutions with different [Cd^2+^]_F_/[thioacetamide]_F_ ratios; (**a**) 4.0, (**b**) 2.0, (**c**) 0.8, (**d**) 0.4.

**Figure 6 nanomaterials-10-00984-f006:**
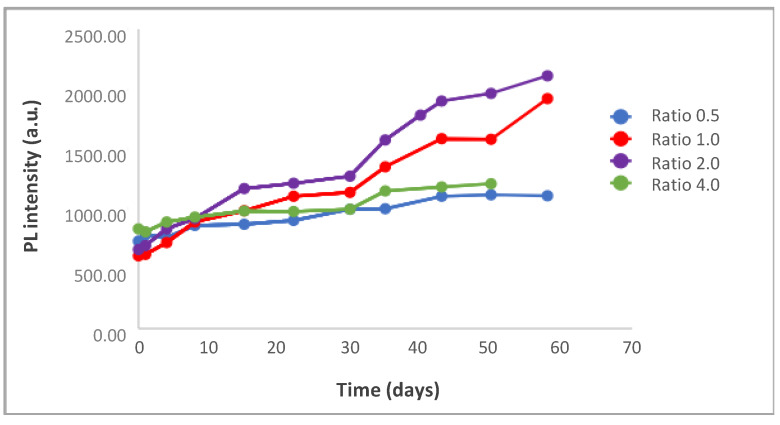
Influence of [Cd2+]_F_/[BSA]_F_ at 4 °C. Temporal evolution of Photoluminescence intensity of CdS QDs solutions with different [Cd^2+^]_F_/[BSA]_F_ ratios.

**Figure 7 nanomaterials-10-00984-f007:**
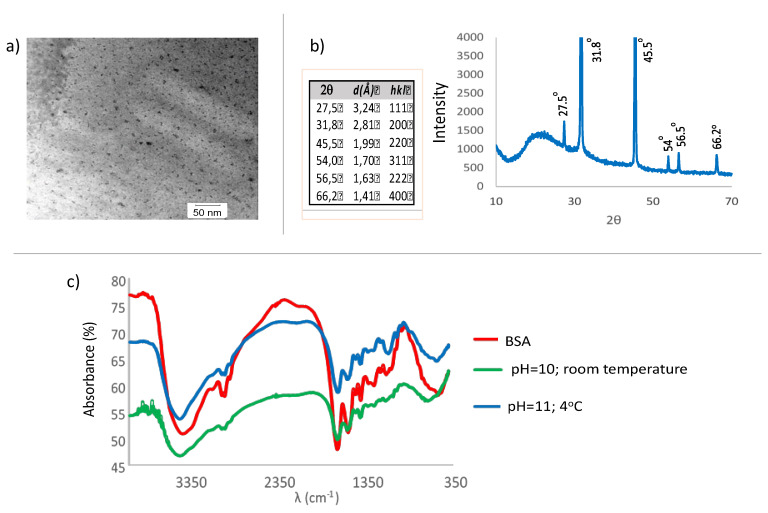
(**a**) Transmission electron microscopy (TEM) image of the CdS nanoparticles; (**b**) 1 indices hkl for the zinc blende structure for the CdS. Powder X-ray diffraction (XRD) pattern of the CdS particles; (**c**) Infrared spectra of the CdS nanoparticles. FT-IR spectra (Fourier Transform InfraRed spectra) of BSA and of CdS QDs prepared at pH 10 (room temperature) and at pH 11 (4 °C).

**Figure 8 nanomaterials-10-00984-f008:**
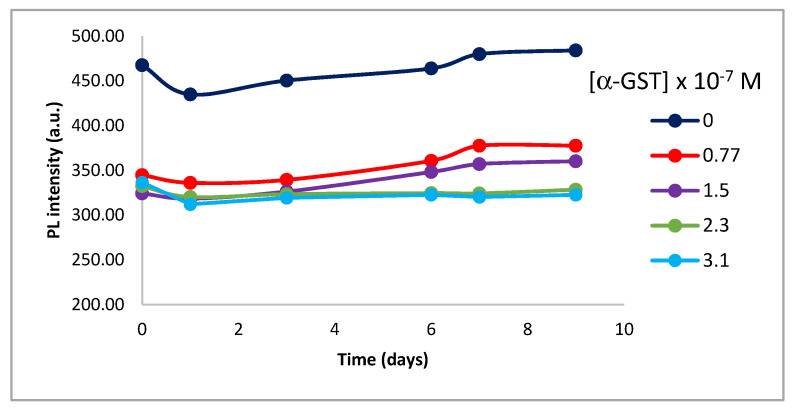
CdS-BSA-α-GST interaction. Variation in fluorescence for the different antibody concentrations studied. Concentration of CdS-BSA: 2.7 × 10^−6^ mol/L. Concentration of α-GST: Range from 7.7 × 10^−8^ mol/L to 3.1 × 10^−7^ mol/L.

**Figure 9 nanomaterials-10-00984-f009:**
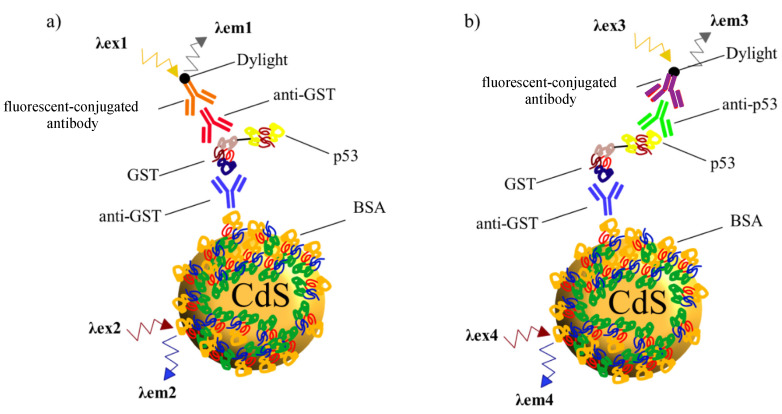
Scheme of detection of in vitro human cell- free transcription-translation (IVTT) expressed human recombinant p53 GST tagged protein. Western Blot detection carried in our study based on novel NPs (CdS-BSA). In both cases, antibody against GST conjugated with the novel QDs (blue), catch the GST domain of the IVTT protein. (**a**) When the goal is detecting the GST domain, another α-GST antibody is used (red) and final detection with a fluorescence dye labeled antibody (orange). (**b**) For specific recognition of the p53 human recombinant protein, an α-p53 antibody is used (green) and detected with a fluorescence dye labeled antibody (purple).

**Figure 10 nanomaterials-10-00984-f010:**
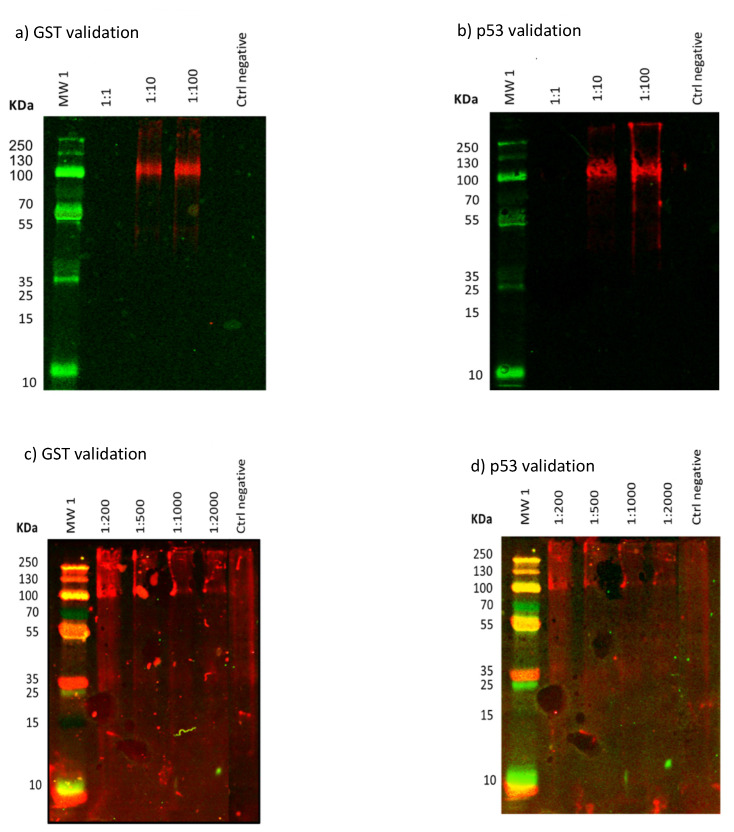
Detection of the GST protein label and the protein under study, p53, after expression of the recombinant protein p53-GST in an in vitro (cell-free) system in the presence of QDs-BSA-α-GST by Western Blot. (In each gel image, lane 1 corresponds to the molecular weight markers, indicating the values (KDa) in the left column; meanwhile the last lane corresponds to negative control). (**a**) Western Blot analysis of IVTT mix with cDNA 1 µg and 5 µL of IVTT reagents (dilution 1:1 at 1:100 *v*/*v*) for the detection of GST at 480 nm. (**b**) Western Blot analysis IVTT mix with cDNA 1 µg and 5 µL of IVTT reagents (dilution 1:1 at 1:100 *v*/*v*) for detection of p53 at 480 nm. (**c**) Western Blot analysis IVTT mix with cDNA 1 µg and 25 µL of IVTT reagents (dilution 1:200 to 1:2.000 *v*/*v*) for the detection of GST at 680 nm. (**d**) Western Blot analysis IVTT mix with cDNA 1 µg and 25 µL of IVTT reagents (dilution 1:200 to 1:2000 *v*/*v*) for the detection of p53 at 680 nm.

**Figure 11 nanomaterials-10-00984-f011:**
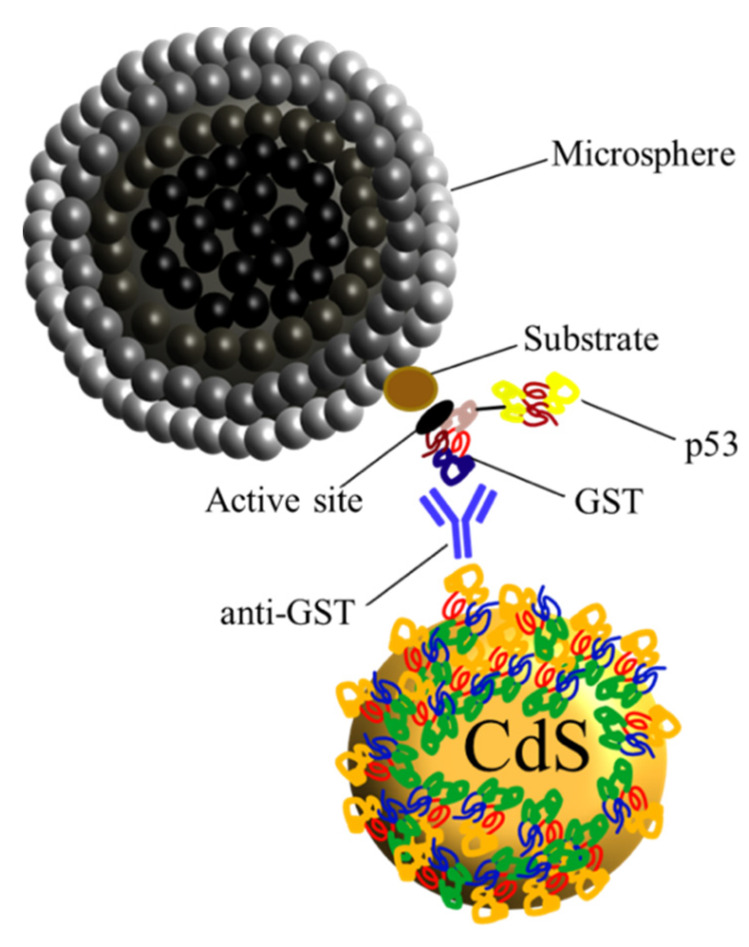
Schematic representation of super-paramagnetic microsphere (black) interaction by a glutathione group (brown), with the active site of the GST enzyme, expressed by IVTT system. New synthetized protein (GST-tagged) is caught by the α-GST antibody (blue) previously conjugated to the novel QD.

**Figure 12 nanomaterials-10-00984-f012:**
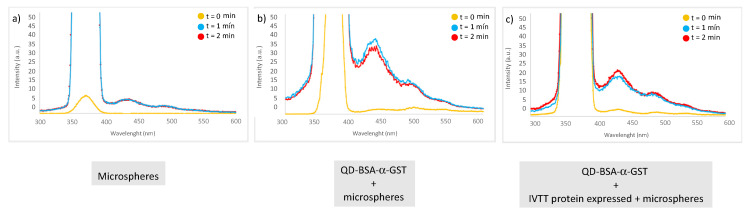
Fluorescence emission spectra at λex = 370 nm, (**a**) microspheres (**b**) QDs-BSA-α-GST and microspheres, (**c**) QDs-BSA-α-GST with expression of p53 and microspheres.

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
