# Peer review of "Detection of Human p53 In-Vitro Expressed in a Transcription-Translation Cell-Free System by a Novel Conjugate Based on Cadmium Sulphide Nanoparticles"

_nanomaterials, 2020, doi:10.3390/nano10050984_

Round 1

Reviewer 1 Report

The manuscript by Barba-Vicente et al. describes...

I think there is some need for english editing because some sentences are difficult to understand.

The abstract contains too many abbreviations not defined, and it is very difficult to read.

The introduction does not end wth the purpose of the work.

In the materials and methods I find some of the descriptions not good enough to allow for repetition. Also, too many abbreviations are not explained at first mention.

The legends to figures are also missing relevant details. The authors must remember that a figure and its legend should be understandable without referring to the whole text.

The results and discussion are highly speculative. And the text is also indicative of this because of the use of "seems", "appears", "probably" and similar expressions. The whole discussion of results has no reference to published works (just 2 papers).

The conclusion do not clarify the practical applications of this system.

The supplementary material has not been cited completely throughout the paper.

Specific comments

L47, L53, L57, L67, L69, L71, L81 there is no use for "suspension points" in a scientific paper

L54-57 this sentence is not complete

L68-71 also this sentence is not complete

L87 the abbreviation is HSA

L93-107 this is a summary of the paper and not the purpose of the work, as should be indicated in the last part of introduction

L107 reference 30 comes before references 26-29

L111 and following - the manufacturer should be indicated with city and country, not website, and it is useless to repeat the same many times in few lines. Group the references together and cite only once

L136 and following - the correct indication of molarity is now mol/L

I am not commenting further the Materials and methods. Too many details are missing, too many protocols are not described.

L223 the Figure contains numbers and abbreviations which are not explained in the legend

L232 Figure 2, I think the two graphs are not correctly labeled, 369.8 is in (a), not in (b)

L239 I do not understand these numbers

L250 the use of "S2-" is highly confusing

L285 in this whole paragraph concerning the synthesis of NP, the discussion makes almost no reference to any other work. A reference to a biochemistry textbook is not appropriate.

L406 the supplementary material tables and figures are not cited in order according to numbers

L450 the "reading pattern" is the "reading frame"

L465 Figure 9 is not understandable without an explanation of the abbreviations

L479 Figure 10 cannot be understood without an indication of the bands and their molecular weight in the different panels

L488-493 this paragraph must be rewritten

L504 Figure 12 should be Figure 11, and it is not understandable without an explanation

L511 wrong figure number. The legend is not understandable

L524 I have seen no sufficient evidence of this

Author Response

We thank the reviewer for the time to read our manuscritp, give opinions and insightful comments which have been included in this updated version of manuscritp. In the enclosed file, it is detailed the specific points required in your review. 

Reviewer 2 Report

The authors report the preparation of CdS QDs in aqueous solution using BSA as capping ligand. The conjugation of CdS@BSA QDs with the α-GST antibody and the biocompatibility of the dots were also investigated. Some of the results presented may be of interest but the manuscript can clearly not be accepted in the present state. From my opinion, the novelty of the work is modest and the paper suffers from the modest characterization of CdS QDs prepared. Moreover, along the whole manuscript results are poorly discussed in the context of literature.

  • the novelty of the work presented must be highlighted.
  • to evaluate the quality of the dots produced, the results should not be discussed using the PL intensity but based on PL quantum yields.
  • line 239: ... intensity at 496 nm of 381.63 (?). Please clarify the text.
  • the quality of the figures must be improved. Labels a, b,... are also missing in some figures.
  • Figure 7a : TEM results must be discussed and a size distribution added.
  • Figure 7c : BSA should be linked to the surface of the dots via a Cd-S bond.
  • Figure 7b: assign the peaks in the XRD pattern. The signals are not broad, which is commonly observed for CdS QDs. These results should be checked.
  • the manuscript contains a few typing errors, see for example wavelength in figure 2, and must be carefully checked by the authors.
  • DLS and zeta potential measurements should be conducted to evaluate the colloidal stability of the QDs prepared.
  • the optical properties and the colloidal stability of the CdS QDs prepared must be compared to those of CdS QDs prepared in aqueous phase.

Author Response

We thank the reviewer for the time to read our manuscript, give opinios and insightful comments with have been included in this updated version of the manuscript. Please, find enclosed a file with detailed info about the specific points requiered in your review.

Reviewer 3 Report

  1. Please explain how changing the pH in such a narrow range from 9 to 10 can affect the change in the stability of nanoparticles with BSA?
  2. To Fig. 3 and Fig.4 is a small clarification: Did you determine the photoluminescence intensity at different pH and temperature values ​​in several parallels?
  3. Fig.10 Figure 10 needs more clarity

Conclusion: Despite the insignificant comments, the article “Detection of human p53 in-vitro expressed in a transcription-translation cell-free system by a novel conjugate based on cadmium sulphide nanoparticles”, Autors: Victor Barba-Vicente , Maria Jesus Almendral Parra * , Juan Francisco Boyero-Benito , Carlota Auria-Soro , Pablo Juanes-Velasco , Alicia Landeira-Viñuela , Alvaro Furones-Cuadrado , Angela-Patricia Hernandez , Raul Manzano-Roman , Manuel Fuentes is extremely relevant, contains the latest original results, has a practically significant direction and can be published in a Journal Nanomaterials

Author Response

We would like to thank the reviewer all your comments and suggestion about our work, which are helping to improve it. According to your comments, here, it is described the changes that we have included in the updated version of our manuscript. 

Round 2

Reviewer 1 Report

The manuscript has been greatly improved and all comments taken into consideration.

Author Response

We thank the reviewer for the time to read our manuscript, give opinions and insightful comments which have improved the final versión of the manuscript.

Reviewer 2 Report

Some of the corrections suggested by the reviewer were done. In the present state, I cannot recommend this manuscript for publication in Nanomaterials.

The language must still be improved and numerous parts of the manuscript require to be clarified/rewritten. The paper contains also many typing errors.

The quality of figures must be improved and their legend clarified. See for example, Figure 7c : FT-IR spectra of BSA and of CdS QDs prepared at pH 10 (room temperature) and at pH 11 (4°C).

The XRD results do not correspond to CdS QDs but to CdS particles.

It has no sense to discuss the results using the PL intensity (see for example, line 258).

Author Response

We thank the reviewer for the time to read our manuscript, give opinions and insightful comments which have improved the final version of the manuscript. 

Reviewer 3 Report

"Detection of human p53 in-vitro expressed in a transcription-translation cell-free system by a novel conjugate based on cadmium sulphide nanoparticles" Authors Victor Barba-Vicente , Maria Jesus Almendral Parra * , Juan Francisco Boyero-Benito , Carlota Auria-Soro , Pablo Juanes-Velasco , Alicia Landeira-Viñuela , Alvaro Furones-Cuadrado , Angela-Patricia Hernandez , Raul Manzano-Roman , Manuel Fuentes  is extremely relevant, contains original results that are obtained by adequate research methods and can be published in Journal Nanomaterials

Author Response

(The authors gave the same response as above.)

Round 3

Reviewer 2 Report

Some of the corrections suggested by the reviewer were done. However, in the present state, it is difficult for me to recommend this manuscript for publication in Nanomaterials.

  • the quality of the figures must still be improved.
  • line 72-75 : clarify the sentence.
  • lines 240, 241 : wavelength
  • line 251 : add a reference
  • line 356 : rewrite the sentence. The mean radius is constant and of ca. 1.5 nm
  • line 429 : correct "sen" into "sin"
  • symbols appear in figure 7b
  • XRD results are not consistent with the small diameters determined for CdS nanoparticles. The authors must compare their results with those of literature.
  • the language must be improved.

Author Response

We thank the reviewer for the time to read our manuscript, give opinions and insightful comments which have been included in this updated versión of the manuscript. Here, it is detailed the specific points required in your review.
